# Robust Imitation of a Few Demonstrations with a Backwards Model

**Jung Yeon Park**
Khoury College of Computer Sciences
Northeastern University
Boston, MA, USA
`park.jungy@northeastern.edu`

**Lawson L.S. Wong**
Khoury College of Computer Sciences
Northeastern University
Boston, MA, USA
`lsw@ccs.neu.edu`

## Abstract

Behavior cloning of expert demonstrations can speed up learning optimal policies in a more sample-efficient way over reinforcement learning. However, the policy cannot extrapolate well to unseen states outside of the demonstration data, creating covariate shift (agent drifting away from demonstrations) and compounding errors. In this work, we tackle this issue by extending the region of attraction around the demonstrations so that the agent can learn how to get back onto the demonstrated trajectories if it veers off-course. We train a generative backwards dynamics model and generate short imagined trajectories from states in the demonstrations. By imitating both demonstrations and these model rollouts, the agent learns the demonstrated paths and how to get back onto these paths. With optimal or near-optimal demonstrations, the learned policy will be both optimal and robust to deviations, with a wider region of attraction. On continuous control domains, we evaluate the robustness when starting from different initial states unseen in the demonstration data. While both our method and other imitation learning baselines can successfully solve the tasks for initial states in the training distribution, our method exhibits considerably more robustness to different initial states.

## 1 Introduction

While reinforcement learning (RL) has shown remarkable success in many challenging domains [21, 38, 36], tasks with sparse rewards and long horizons still remain extremely difficult to solve. In such tasks, a positive reward is only encountered after the RL agent reaches the goal after a long sequence of actions, meaning that it cannot learn any useful signals until this occurs (typically the agent must reach the goal several times to learn reliably as well). Furthermore, the learning signal decreases exponentially with the horizon, which combined with slow gradient-based updates, can lead to catastrophic forgetting even after the agent learns to reach the goal.

Expert demonstrations can help RL agents solve difficult tasks [30, 43, 22]. These demonstrations can be used in the supervised setting where the agent imitates the expert's behavior, termed imitation learning (IL). However, naive behavior cloning (BC) of the expert's trajectories suffers from covariate shift: the agent's policy drifts away from the expert's, which leads to compounding errors due to RL's sequential nature. Furthermore, the distribution of states given in the demonstrations often has low-dimensional support with respect to the entire state space. As such, the agent cannot extrapolate correctly when outside of the demonstration data. Many approaches to solve this issue have been proposed [33, 39, 16], but such approaches require an interactive expert to query the correct actions.

Another way to use demonstrations is to combine imitation learning with reinforcement learning in a form of learning from demonstrations (LfD) [35, 13, 9]. In this case, demonstrations do not simply act as supervised labels and can guide the agent's exploration, and also act as augmentations to good

36th Conference on Neural Information Processing Systems (NeurIPS 2022).

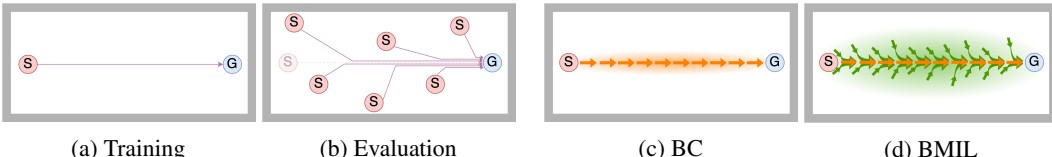

| (a) Training | (b) Evaluation | (c) BC | (d) BMIL |

Figure 1: **(a)**, **(b)**: Robustness: The policy is trained on a specified set of initial start and goal states and is evaluated at different start states. **(c)**, **(d)**: BMIL uses generated reverse-time rollouts from a backwards model (green arrows originating from the demonstration) to learn a wider region of attraction (green) around the demonstration data (orange) than BC.

data samples. These LfD approaches either use demonstrations to pretrain the policy [35, 9], use an auxiliary imitation loss in conjunction with the policy update [30, 22], or modify the reward function such that the agent is rewarded when it imitates the demonstrations [50, 31]. However, these methods require interactions with the environment whereas we do not assume such access in our setting.

One primary concern of relying on demonstrations is that they are costly to obtain, especially in real-world applications. Requiring an operator providing corrections in-the-loop to handle covariate shift is often prohibitive as well. Given only a few offline trajectories demonstrating successful task completion, an agent ought to be able to replicate the behavior from similar starting conditions, even if there are small perturbations along the way, and correct itself when necessary. We are primarily interested in this setting. In this work, we aim to minimize the number of demonstrations necessary for sparse-reward tasks, while preserving successful task completion. To tackle covariate shift, we seek an approach that will be *robust*, in the sense of Figures 1a and 1b: the agent is only trained with a few demonstrations from a single start state, but at evaluation time, it must generalize its behavior to a variety of unseen start states. If the agent can successfully complete the task from states adjacent to the demonstrated trajectory, then it can recover from small deviations from the demonstration.

Inspired by the notion of "funnels" in robotics [20] and feedback control [2, 19], we introduce a reverse-time generative model that can generate possible paths leading the agent back onto the demonstrations. These reverse rollouts provide useful information because every rollout ends within the support of the demonstration data. Assuming that the demonstrations lead to the goal, imitating these generated reverse rollouts and the original demonstration data allow the agent to reach the goal from more starting conditions, including unseen ones. As illustrated in Figures 1c and 1d, typical behavior-cloning (BC) methods focus learning on the small number of demonstrated states, whereas our proposed approach, Backwards Model-based Imitation Learning (BMIL), uses the reverse rollouts to learn a wider region of attraction around the demonstration. We validate our approach on a number of long-horizon, sparse-reward continuous-control tasks. Even from 5–20 demonstrations, BMIL provides a significant increase in the region of attraction and robustness on many domains compared to BC, or to using a forward dynamics model.

We summarize our contributions as follows:

- We propose an imitation learning method that pairs a backwards dynamics model with a policy and train on both demonstrations and imagined model rollouts.
- In the restrictive setting of an offline expert and no access to environment interactions, we show that a backwards model can improve robustness over behavior cloning.
- Our experiments on a variety of long-horizon, sparse-reward domains demonstrate that BMIL can noticeably extend the region of attraction around the demonstration data, even when trained on very small subsets of the state space.

## 2   Related Work

Imitation learning has a long history [27, 35] and is well-studied, as documented in various surveys [1, 25, 11]. The challenges of covariate shift and compounding errors are also well-known [27, 32]. Most solutions involve on-policy imitation learning, where environment interaction and interactive querying of the expert allow for the agent's distribution to match that of the expert [33, 39]. More closely related to our approach are methods that modify or augment the demonstrations to increase robustness. Laskey et al. [16] inject noise into the supervisor's policy during training to force the

demonstrator to provide corrections. Luo et al. [18] learn a dynamics model from demonstrations to conservatively extrapolate a value function that encourages the agent to return to the expert data distribution. Generative approaches have also been used in imitation learning [10, 46], but their focus is not on robustly following a few goal-reaching demonstrations.

Time-reversibility has been explored in RL, often as a form of regularization [41, 23, 49, 29, 34]. Reverse-time dynamics models, also called predecessor models, have also been used in RL [3, 6, 37, 15, 17, 48, 7]. However, in all cases, the reverse-time dynamics model is used as either an alternative to the forward-time dynamics model or as an auxiliary model in addition to the standard forward-time dynamics model, in order to mitigate model-compounding errors. The result is that the reverse-time dynamics model can accelerate RL and enable greater sample-efficiency. In this work, we take a different perspective where the reverse-time dynamics model is used to generate possible, unseen paths that can lead back to the expert demonstration and thus to the goal, thereby improving robustness in following the demonstration. Our work is also similar to [45], where a reverse-time dynamics model is used to generate possible trajectories; however, their focus is on offline RL, where the generated trajectories are used to connect distinct sets of states in the offline dataset.

# 3 Method

## 3.1 Preliminaries

We model the setting as a Markov Decision Process (MDP) with a continuous state space $\mathcal{S}$ and a continuous action space $\mathcal{A}$. $T : \mathcal{S} \times \mathcal{A} \mapsto \mathcal{S}$ is the transition function that defines the distributions of the next state $s_{t+1} \in \mathcal{S}$, given the current state $s_t \in \mathcal{S}$ and action $a_t \in \mathcal{A}$ taken at timestep $t$.

The objective of imitation learning is to learn a policy $\pi_\theta$, parameterized by $\theta$, that matches the expert's policy $\pi_E$. We assume that the expert generates demonstrations $\mathcal{D}_{demo}$ by rolling out its policy $\pi_E$ in the environment. Note that we consider the more restrictive case of where demonstrations only consist of transitions $(s, a, s')$ and not rewards, where $s'$ is the next state.

Behavior cloning (BC) is a form of imitation learning where the policy $\pi_\theta$ learns to imitate expert actions via supervised learning. The policy learned by behavior cloning is found by minimizing the negative log-likelihood over the demonstration data

$$\mathcal{L}_{BC} = \mathbb{E}_{(s,a) \in \mathcal{D}_{demo}} \left[ - \log \pi_\theta(a \mid s) \right].$$

Note that BC does not require environment interactions and can be considered to be offline. Furthermore, expert behavior is inferred only from demonstrations without access to the expert policy.

**Compounding errors in behavior cloning**  As pointed out in [32, 44], behavior cloning suffers from covariate shift, where errors in the policy can compound and lead the agent to states where it cannot recover. Intuitively, this occurs as states in the training data $\mathcal{D}_{demo}$ is a small subset of the entire state space $\mathcal{S}$ and it is difficult for the policy $\pi_\theta$ to learn the optimal action for states outside of the training data. If during a rollout, once the policy $\pi_\theta$ makes an error and leaves the $\mathcal{D}_{demo}$, it may encounter completely new states, leading to the compounding of errors. Furthermore, as the agent moves farther away from $\mathcal{D}_{demo}$, there is very little hope for it to make the correct action and move back onto the distribution of the training data.

## 3.2 Problem Setting

We consider the same setting as BC, where we do not assume access to the environment or the expert policy during training, and only expert demonstrations without rewards are given. Furthermore, we assume that the expert demonstrations are given in the MDP where the sets of initial states $S_0$ and goals $\mathcal{G}$ are both very small subsets of the entire state space. An example of this scenario is a maze environment where the agent starts from the same initial state and tries to reach a fixed goal. Formally, we define the demonstration trajectories $\tau_{demo}$ as coming from a probability density

$$p(\tau_{demo} \mid \pi_E) = p(s_0) \prod_{t=0}^{T-1} \pi_E(s_t \mid a_t) p(s_{t+1} \mid s_t, a_t),$$

where $s_0 \in S_0$, and $s_T \in \mathcal{G}$. In our experiments, $S_0$ and $\mathcal{G}$ consist of a single start or goal state and/or the $\varepsilon$-ball of its neighborhood. As such, only a few demonstrations are required to learn a stable

optimal policy. Note that this is different from domains in previous work [30, 4] which consider random goal states, requiring many more demonstrations to learn optimal policies. We also assume that the expert policy is optimal in the sense that all demonstrations successfully reach a goal. While this is not strictly necessary in our method, our setting does not include rewards in $\tau_{demo}$ and thus we cannot discern whether demonstrations are optimal. This allows us to ignore the issue of modifying rewards as done in several offline RL algorithms [5, 14]. In order to use task completion success rates as an evaluation metric in our experiments, we consider only optimal demonstrations.

Our objective is to learn a policy that is robust to policy errors when imitating expert behavior and can learn to reach the goal from a variety of initial states. This is different than the multi-goal or multi-task setting, where the agent learns to solve multiple goals or tasks, usually from a small number of initial states. More formally, the robustness of the policy $R(\pi_\theta)$ is defined as

$$R(\pi_\theta) = \mathbb{E}_{s_0 \in S_R} \left[ \mathbb{1}\{\exists t \leq T, s_t \in \mathcal{G}\} \right],$$

where the expectation is taken over $S_R$, where $S_R$ is a strict superset of $S_0$. Note that our measure of robustness is somewhat coarse, in that we do not consider the shortest path to reach the goal from every start state (which would probably require more information such as diverse trajectories or environment interactions). Instead, we seek to extend the region of attraction around $\mathcal{D}_{demo}$ such that the learned policy can still reach the goal.

As we consider continuous states in our work, we measure the robustness $R$ using samples from $S_R$, where we randomize either some or all of the state dimensions. For example, in robotic manipulation domains, we vary the position of the gripper as we are primarily concerned with being able to learn robustness from a variety of different starting positions. In other domains, we are interested in the policy's ability to recover from arbitrary initial states and so we vary not only the agent's starting position, but also the initial joint positions and velocities by adding uniformly random noise.

Throughout, we assume that there exists an action $a \in \mathcal{A}$ that allows the policy to go towards $\mathcal{D}_{demo}$ when in a state $s \notin \mathcal{D}_{demo}$. This scenario is true for many navigation and physics-based domains if we ignore rare circumstances such as irrecoverable unsafe states or the breakdown of the agent. We exclude such cases and assume that state transitions are reversible. We discuss some possible ways to incorporate irrecoverable states in Section 6.

### 3.3 Backwards Model-based Imitation Learning

In our work, we use a backwards dynamics model to provide more synthetic training data to the policy and therefore increase the policy's robustness. We call our method backwards model-based imitation learning or BMIL.

**Backwards model**  The backwards model is a probabilistic generative model defined as $B = p(s_t, a_t \mid s_{t+1})$. This model estimates the conditional distribution of the reverse time dynamics. It takes in the next state and outputs the previous state and previous action. As we consider only continuous state and action spaces, we implement $B$ as a conditional Gaussian, parameterized by $\phi$.

The backwards model is decomposed into two functions $B = B_A \cdot B_S = p(a_t \mid s_{t+1}) \cdot p(s_t \mid a_t, s_{t+1})$, an action generator and previous state generator. The action generator $B_A$ predicts which action was taken in order to land in the next state. There may be several such actions from different states that can lead to the next state. Thus the action generator implicitly encodes a backwards policy. It is important for this backwards policy to closely match the learned forward policy $\pi$ but be different enough to generate diverse new rollouts for the policy to train on. The previous state generator $B_S$ predicts the previous state given the next state and previous action taken. The goal of this generator is to accurately predict the backwards dynamics.

As we consider the setting with no access to the expert or the environment, $B$ is trained only on $\mathcal{D}_{demo}$. The action generator and previous state generator are jointly trained by maximum likelihood

$$\mathcal{L}_B = \sum_{t=0}^{T} \log p(a_t \mid s_{t+1}) + \log p(s_t \mid a_t, s_{t+1}), \tag{1}$$

where $s_t, a_t, s_{t+1}$ is the state, action, and next state, respectively, at timestep $t$.

**Model rollouts**  Given expert demonstrations $\tau_{demo}$, we use the backwards model to generate possible several short reverse rollouts or traces $\tau_B$, starting from every state in $\tau_{demo}$. As all of these

**Algorithm 1** Backwards Model-based Imitation Learning (BMIL)

1: **for** $N$ epochs **do**
2:     Train backwards model parameters $\phi$ using Eqn. (1).
3:     Generate $K$ model traces $\tau_B$ from every state $s \in \mathcal{D}_{demo}$ and store them in $\mathcal{D}_M$.
4:     **for** $M$ steps **do**
5:         Sample mini-batch of $(s, a)$ from $\mathcal{D}_{demo}$ and $\mathcal{D}_M$ at a fixed ratio.
6:         Update policy parameters $\theta$ using Eqn. (2).
7:     **end for**
8: **end for**

traces end on states within the demonstration data $s \in \mathcal{D}_{demo}$ and as all demonstrations reach the goal, following these traces will eventually lead to the goal. For all $s_1, \ldots, s_T$ in $\tau_{demo}$, we generate $K$ traces $\tau_B$ in a time-reversed manner, where we start from the last state action pair $(s_H, a_H) \in \mathcal{D}_{demo}$ and then predict $(s_t, a_t)$ for timesteps $t = H-1, \ldots, 1$. These traces are collected into a buffer $\mathcal{D}_M$. As we assumed that there are no irrecoverable states in our setting, the rollouts reflect possible paths that the agent could have taken to reach $s \in \mathcal{D}_{demo}$. If the reverse time model $B$ is accurate and the previous action generator $B_A$ gives sufficiently diverse actions, the traces $\tau_B$ are then samples from the region of attraction or "funnels" around every state along the demonstration. As we assume all demonstrations reach the goal, these samples from the funnels can be used to learn a robust policy $\pi_\theta$, as it can follow the traces onto the optimal path.

**Action selection strategy for** $B_A$    As the backwards model $B$ is trained only on a limited number of expert demonstrations, it is likely that $B$ can only learn accurate reverse-time dynamics for states contained within or close to the demonstration data $\mathcal{D}_{demo}$ [47]. Thus repeatedly rolling out $B_A$ would only generate traces whose state-action pairs are contained within $\mathcal{D}_{demo}$ and would not help with learning robust policies. However, we would like to generate diverse traces with new unseen state-action pairs in order to robustify the policy. To balance the model misprediction accuracy with generating plausible state-action pairs, we perturb only the first action generated from $B_A$ and not the subsequent actions and also use short horizon lengths for the traces. Note that we are essentially choosing a good action selection strategy for $B_A$. Let $a_E$ be the action that the expert would take. A good action selection strategy would place more probability mass closer to the support of the $\mathcal{D}_{demo}$, providing a "cover" of $p(a_E|s)$ but with a wider tail to provide diverse rollouts. As our backwards model $B$ is probabilistic (implemented as a conditional Gaussian with diagonal covariance), we can easily perturb $p(a_E|s)$ by increasing the distribution's variance.

Let $a \sim \mathcal{N}(\mu, \sigma^2)$ be the previous action output by $B_A$. We consider two ways to generate action $a$: 1) simple scaling of $\sigma$ and 2) resampling a new action $a'$ by adding uniform noise, $a' = a + k, k = \mathcal{U}[-b, b]$, where $b$ is a fixed hyperparameter. For the scaling strategy, we further scale $\sigma$ by the entropy of the probability density as we wish to make the distribution "wider" for peaker distributions.

**Algorithm**    Our method BMIL is outlined in Algorithm 1. Given expert demonstrations with tuples of the form $(s_t, a_t, s_{t+1})$, we train the backwards model using Eqn. 1 to estimate the reverse-time dynamics $p(s_t, a_t \mid s_{t+1})$. We train our policy $\pi_\theta$ on both the demonstration data $\mathcal{D}_{demo}$ and the model traces $\tau_B$ by sampling from both at a fixed ratio and using maximum likelihood,

$$\mathcal{L} = p_d \mathcal{L}_{BC} + (1 - p_d)\mathbb{E}_{(s,a)\sim\tau_B} \left[-\log \pi_\theta(a \mid s)\right], \qquad (2)$$

where $p_d$ is the probability of sampling from $\mathcal{D}_{demo}$. As our aim is to learn a robust policy while still succeeding at the original start states and goals, we sample from the demonstrations at a higher ratio than the model traces. Note that BMIL does not depend on the type of imitation learning policy. Any algorithm can be used as long as the policy can be trained with samples of the form $(s, a)$.

## 4 Experiment Design

### 4.1 Environments

We validate our approach on several continuous control domains: 1) the Fetch robotics environment [26], 2) maze navigation with two different agents, and 3) Adroit hand manipulation [30]. Figure 2 shows sample images of some environments. For the Fetch robotics environments, we consider the

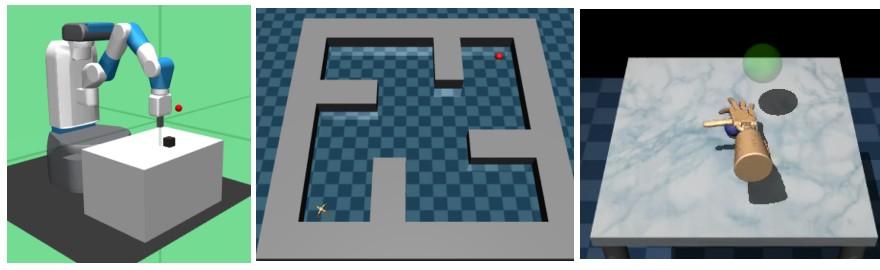

| Fetch-PickAndPlace | Maze-AntCorridor7x7 | Adroit-Relocate |

Figure 2: Sample images of some considered environments.

"Push" and "PickAndPlace" tasks where the objective is to control a Fetch end effector to either push an object to the goal or pick an object and place it at the target location. For the maze environments, we consider 3 mazes of increasing difficulty, where an agent must learn to move itself and then reach the goal. We use both a simple point and a 29-DoF ant agent. For the Adroit environment, we use the 'Relocate' task, where one must control a 24-DoF Adroit hand to pick up a ball and move it to a target location. All domains use the MuJoCo simulator [42] for a total of 9 distinct domains. All environments have sparse reward structures, where either every step has a constant negative reward until the goal is reached (Fetch) or only the goal has a non-zero reward (Maze, Adroit). In particular, the Maze and Adroit environments are quite challenging as they both consist of controlling the agent's joints to perform locomotion (maze) or dexterous manipulation (Adroit) over a long horizon. More detailed descriptions of each environment including its observation space are provided in Appendix A.

## 4.2 Demonstrations and Implementation Details

To generate demonstrations in the Fetch and Maze domains, we train an expert policy by adding the goal position to the state, as in goal-oriented learning, and use off-the-shelf RL algorithms [28, 8]. For the Adroit domain, we use a pre-trained policy from [30]. We use 5 demonstrations on the Push task and 10 on the PickAndPlace task, and 20 demonstrations for all Maze and Adroit environments.

For the policy, we use neural networks with 3 fully connected hidden layers with 256 neurons and ReLU activations. For the backwards model, we use 4-layer MLPs with 256 hidden units for both the action predictor $B_A$ and previous state predictor $B_S$ and use diagonal Gaussian distributions. To train the policy, we use $p_d = 0.5$ for the Fetch environments, $p_d = 0.8 \sim 0.95$ for the Maze environments, and $p_d = 0.8$ for the Adroit environments. We find that higher ratios are necessary for longer-horizon and more complex domains. For the model rollouts, we use the variance scaling action selection strategy for the first action only and use increasing rollout lengths for all domains, similar to [12]. For a more detailed discussion of experiment details, see Appendix B. Our code for the modified environments, generating expert policies, and running all experiments are available at `https://github.com/jypark0/bmil`.

## 4.3 Evaluation

We evaluate BMIL against behavior cloning (BC) and VINS [18]. VINS specifically aims to learn value functions robust to perturbations using negative sampling and the induced policy learns self-corrective behavior. VINS was chosen as it is most relevant to our setting; other methods such as DART [16] or SQIL [31] require either an online expert or environment interactions.

We use the same number of demonstrations for all methods and also keep the same policy network architecture and the total number of policy gradient steps equal across all methods. We train both the policy and backwards model until the backwards model loss converges. Note that our goal is not to solve the training task faster but rather to robustify the policy using the backwards model. Additionally, we wish to solve the task at various starting conditions while still being able to succeed at the original initial start-goal states.

To evaluate the robustness of the learned policy, we vary the initial states and compute task success rates. For Fetch, we fix the initial gripper, object, and goal position during training and vary the gripper's $x$ and $y$ position within the table boundaries during evaluation. We use 10,000 samples

| | | | Robustness (%) | | | Relative to BC | | |
|---|---|---|---|---|---|---|---|---|
| | | | BC | VINS | BMIL | BC | VINS | BMIL |
| FETCH | Push (5 demos) | | $12.1_{\pm 0.3}$ | $12.8_{\pm 0.4}$ | $\mathbf{14.6}_{\pm 0.6}$ | 1 | 1.06 | **1.21** |
| | PickAndPlace (10 demos) | | $4.1_{\pm 0.1}$ | $3.4_{\pm 0.1}$ | $\mathbf{17.5}_{\pm 0.9}$ | 1 | 0.84 | **4.31** |
| MAZE | Point (20 demos) | UMaze | $\mathbf{49.0}_{\pm 1.9}$ | $39.5_{\pm 2.1}$ | $47.8_{\pm 3.5}$ | 1 | 0.81 | 0.98 |
| | | Room5x11 | $36.8_{\pm 3.4}$ | $17.3_{\pm 2.8}$ | $\mathbf{38.6}_{\pm 3.4}$ | 1 | 0.47 | **1.05** |
| | | Corridor7x7 | $33.7_{\pm 1.5}$ | $\mathbf{37.7}_{\pm 1.2}$ | $\mathbf{38.9}_{\pm 2.3}$ | 1 | **1.12** | 1.16 |
| | Ant (20 demos) | UMaze | $\mathbf{63.0}_{\pm 1.0}$ | $44.7_{\pm 2.1}$ | $\mathbf{64.8}_{\pm 1.5}$ | 1 | 0.71 | 1.03 |
| | | Room5x11 | $\mathbf{33.2}_{\pm 0.9}$ | $30.2_{\pm 0.8}$ | $29.1_{\pm 0.8}$ | 1 | 0.91 | 0.87 |
| | | Corridor7x7 | $\mathbf{21.7}_{\pm 0.6}$ | $19.6_{\pm 0.6}$ | $17.6_{\pm 0.5}$ | 1 | 0.90 | 0.81 |
| ADROIT | Relocate (20 demos) | | $7.9_{\pm 0.7}$ | $3.8_{\pm 0.7}$ | $\mathbf{13.3}_{\pm 1.0}$ | 1 | 0.48 | **1.68** |

Table 1: Robustness evaluation for Fetch, Maze, and Adroit environments over $400, 100, 100$ trials, respectively. The bounds indicate 95% confidence intervals. BMIL improves robustness considerably over BC in most environments.

during evaluation. For Maze, we initialize the agent to a random start position within a discretized grid of the maze and also add random uniform noise to the agent joints' qpos,qvel. We sample 100 initial states for each discrete grid cell and compute the success rate. Sampling points for each grid cell gives us an idea of which positions are easy for the agent to reach the goal. Intuitively, such positions would be those near the goal and the demonstrated path. For Adroit, we generate $1,000$ random initial states by adding uniform noise to the qpos of the hand.

## 5 Results

Our experiments aim to answer the following questions: 1) how robust of a policy does BMIL learn? and 2) what components of BMIL are important to improve robustness?

### 5.1 Robustness evaluation

The robustness results are shown in Table 1. We note that the absolute robustness percentages are generally low for all methods because of the difficulty in extrapolating from limited demonstrations $(5 - 20)$ with a single pair of initial start and goal states. We therefore also include the relative improvement over BC.

In the Fetch environments, BMIL substantially increases robustness over BC and VINS. In particular, our method has an approximately $20\%$ and $330\%$ improvement over BC on Push and PickAndPlace, respectively. VINS on the other hand performs similarly to BC. We see a similar pattern on the harder Adroit environment, where BMIL improves robustness over BC by $68\%$. For the Maze environments, BMIL generally outperforms BC for the Point agent, while the robustness is decreased for the Ant agent. Somewhat surprisingly, BC performs quite well on the long-horizon Maze domains. It may be that BC has some built-in extrapolation capabilities or that the backwards model may need better latent representations with more powerful networks.

Empirically, we can see that having short reverse rollouts from the backwards model and using only slight perturbations still helps to increase robustness, even though the traces contain some model misprediction errors. We hypothesize that these traces do not necessarily need to be accurate in order to benefit the policy and simply need to be plausible paths that lead to the demonstrations. It may be that having the general correct direction contained in the traces is sufficient for the policy to eventually reach states in the demonstration data.

The success rates during training are shown in Table 5 in Appendix C.1. BMIL achieves success rates close to $1$ for most domains, suggesting that increased robustness does not necessarily come at the cost of decreased performance on demonstrated start-goal states. On the other hand, VINS cannot consistently succeed during training, even though its robustness is similar to BC.

**Visualization of robustness**  Figure 3 shows which starting positions succeed during the robustness evaluation for Fetch. The green points correspond to successful episodes. We can see that both BC and BMIL succeed more frequently when starting from nearby the demonstration data (approximately

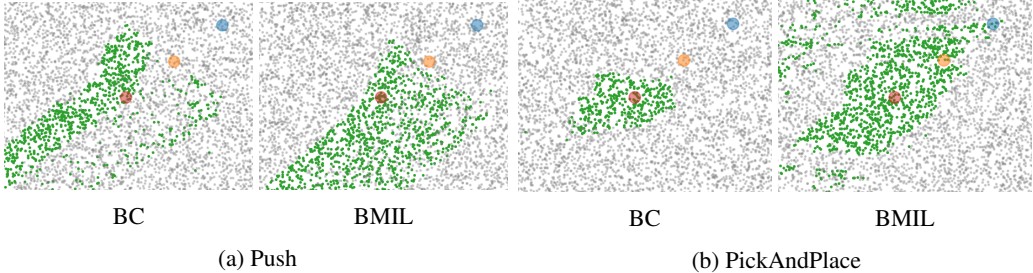

|          |          |          |          |
|:--------:|:--------:|:--------:|:--------:|
| BC       | BMIL     | BC       | BMIL     |
| (a) Push |          | (b) PickAndPlace |  |

Figure 3: Visualization of robustness on Fetch for BC and BMIL. We vary the gripper's $x, y$ position to evaluate robustness. The green and gray points denote successful and unsuccessful initial positions. The red, orange, and blue points denote the initial start, object, and goal during training. BMIL learns a much larger region of attraction along the path from the initial start (red) to the goal position (blue) and even succeeds in some other areas much farther away.

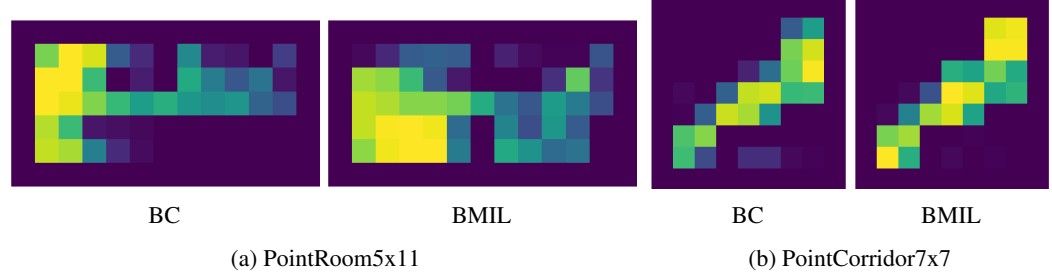

|          |          |          |          |
|:--------:|:--------:|:--------:|:--------:|
| BC       | BMIL     | BC       | BMIL     |
| (a) PointRoom5x11 |  | (b) PointCorridor7x7 |  |

Figure 4: Visualization of robustness on PointRoom5x11 and PointCorridor7x7 for BC and BMIL. The maze is discretized into a grid and 100 random states are sampled from each grid positions (each random state also adds noise to the agent's initial joint positions/velocities). Bright yellow corresponds to 100% success rate while dark purple corresponds to 0%. The regions of attraction for both BC and BMIL are similar but BMIL succeeds more often within its region of attraction.

a straight line from the start (red) to goal (blue)). However, we can see that BMIL learns a much larger region of attraction over BC and even succeeds at points that are much farther away from $\mathcal{D}_{demo}$. We hypothesize that instead of perturbing a single state within $\mathcal{D}_{demo}$ as done in VINS, learning a short reverse rollout from this state allows BMIL to learn optimal paths from states much farther away from $\mathcal{D}_{demo}$, leading to higher robustness values. Figure 4 shows a similar visualization for some Point maze environments where the agent's position is discretized into a grid. BMIL learns a region of attraction that is either slightly bigger or similar in size to that of BC but has a higher rate of success at each cell.

## 5.2 Additional Experiments

**Forward vs Backward Dynamics** We first analyze the utility of a backwards vs a forward dynamics model. On the Fetch environments, we train BMIL with a forward dynamics model $p(s' \mid s, a)$ and compare against the original backwards model $p(s, a \mid s')$. The forward model is implemented nearly identically to other n-step model-based RL algorithms (e.g. [12]), with the exception of no environment interactions. As with the backwards model, we generate rollouts from the forwards model starting from demonstrated states and train the policy on both the demonstrations and traces.

|  | Robustness (%) | | | Relative to BC | | |
|---|---|---|---|---|---|---|
|  | BC | BMIL (Forwards) | BMIL (Backwards) | BC | BMIL (Forwards) | BMIL (Backwards) |
| Push (5 demos) | $12.1_{\pm 0.3}$ | $12.4_{\pm 0.6}$ | $14.6_{\pm 0.6}$ | 1 | 1.03 | 1.21 |
| PickAndPlace (10 demos) | $4.1_{\pm 0.1}$ | $4.1_{\pm 0.2}$ | $17.5_{\pm 0.9}$ | 1 | 1.03 | 4.31 |

Table 2: Forwards ($p(s' \mid s, a)$) vs Backwards ($p(s, a \mid s')$) dynamics model: The forwards dynamics model performs similarly to BC and does not increase robustness.

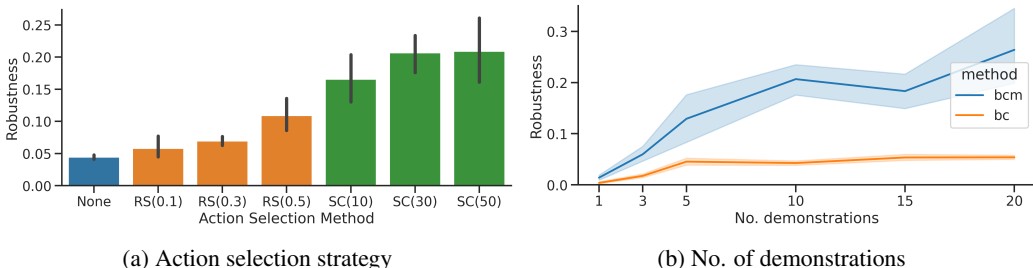

(a) Action selection strategy

(b) No. of demonstrations

Figure 5: **(a)** Action selection strategy: effect of no perturbation (None), resampling (RS), and scaling (SC) on robustness on PickAndPlace. The numbers in the brackets indicate coefficients. We see that perturbing the first action is beneficial compared to the no perturbation (None) method and allows the backwards model to generate diverse traces. **(b)** Number of demonstrations on PickAndPlace: more demonstrations increase robustness for BC and BMIL, but BC plateaus at a much lower level.

To generate model rollouts from demonstration states, we use the action from the policy $a = \pi_\theta(s)$. The total number of parameters is kept approximately constant for the forwards and backwards models. As shown in Table 2, the forwards model offers little to no benefit over BC for both Push and PickAndPlace, suggesting that the backwards model is required to produce a robust policy.

**Action selection strategy** We test different action selection strategies in trace generation in Figure 5a (and Figure 10 in Appendix C.2). Compared to no perturbation (None), we see that either action selection strategy improves robustness as it can lead to more diverse trajectories unseen within the support of the demonstrations. We use the variance scaling strategy SC(30) for all Fetch experiments as it was more stable than SC(50).

**Number of demonstrations** We also study our method's performance with varying numbers of demonstrations. As shown in Figure 5b, both BC and BMIL improve in robustness with more demonstrations, but BC plateaus at a much lower level. On the other hand, BMIL requires slightly more (10) demonstrations than needed (3 demonstrations are sufficient for BC to succeed during training) in order to train the backwards model (Figure 11 in Appendix C.2).

**Computation budget** As BMIL trains both the policy and the backwards model, it requires more total gradient updates than BC. On Fetch domains, BMIL uses approximately 5x more computation than BC. We train BC for more steps to match or exceed BMIL's computation budget, as shown in Figure 6 (BC is given 1x–20x computation budget). However, more training for BC does not improve robustness and seems to have a harmful effect on robustness.

**Training model first and then the policy** As an offline method, BMIL does not require the backwards model to be trained in a single loop along with the policy. We can first train the model first and then train the policy. We compare training the model first and then the policy with the process outlined in Algorithm 1. We find there are no noticeable differences when using this model first approach on the Fetch domains, as seen in Table 6.

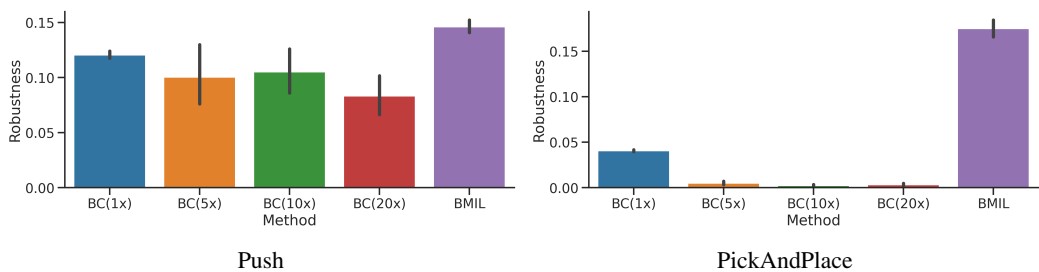

Push

PickAndPlace

Figure 6: Computation budget: we increase the number of gradient steps for BC by $1 - 20x$ for Fetch. BMIL has roughly $5$ times more total gradient steps than BC(1x) due to the backwards model update. More BC gradient steps do not increase robustness.

# 6 Discussion

This work proposes a method to tackle the issue of covariate shift in imitation learning. We consider the restrictive setting where the expert is offline, where its behavior can only be inferred from demonstrations, and no access to additional environment interactions. Specifically, we show that pairing a generative backwards model with behavior cloning can allow a policy to learn a wider region of attraction around the demonstration data. By rolling out imagined traces from states within the demonstration and perturbing actions to generate diverse traces, BMIL learns a wider funnel than naive BC. Through experiments on several long-horizon, sparse-reward, continuous control domains, BMIL noticeably improves robustness when trained on a narrow set of initial start and goal states and evaluated at random starting positions.

There are many possible extensions for future work. BMIL does not necessarily preclude the use of image observations as we only assume that slightly perturbing an action will lead to new next states close to the original next state. However, to handle images, our approach likely requires an additional encoder and possibly more complex network architectures and augmentation techniques. Another interesting avenue could be to quantify how an increasing coverage of state space contained within the demonstration data affects robustness for both BC and BMIL. Finally, one could consider the setting of irrecoverable states and either resample rollouts containing such unsafe states or incorporate a measure of safety within the backwards model when generating model rollouts.

## Acknowledgments and Disclosure of Funding

This material is based upon work supported by the National Science Foundation under Grant No. 2107256. This work was completed in part using the Discovery cluster, supported by Northeastern University's Research Computing team.

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
