# A Environments

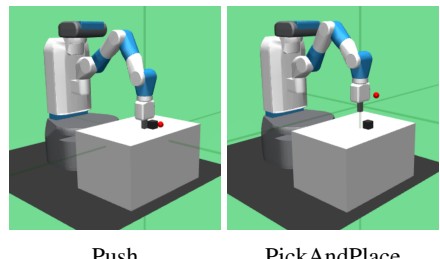

Push          PickAndPlace

Figure 7: Fetch robotics environments. The gripper must move the object (black) near the goal (red).

**Fetch** These environments from [26] involve controlling the Fetch robotic arm. We consider two tasks: "Push" and "PickAndPlace" as shown in Figure 7. The objective of "Push" is to push the object on a table towards a fixed goal position using a closed gripper. The objective of "PickAndPlace" is to pick the object by controlling the gripper and move it towards the goal. The observation space is 25-dimensional and consists of the end effector coordinates and its linear velocity, the gripper's position and velocity, and the object's pose, velocities, and its relative position/velocity to the gripper. All Fetch environments have a fixed episode length of 50 and an episode is considered successful if the object is at the goal at the end of the episode.

**Maze** There are 3 different maze environments of increasing difficulty and 2 different agents (Point, Ant) for each environment (c.f. Figure 8). The initial start and goal position is fixed and the objective is to control the agent to reach the goal (colored in red). The observation space is the agent's joint positions/velocities, the current timestep, and the agent's current Cartesian position. We use $dt = 0.02$ within the Mujoco [42] simulator and set the gear ratio for Ant to 10 to prevent it from falling over as in [24].

**Adroit** In this domain from [30], the goal is to control a 24-DoF Adroit hand to pick up a ball from the table and move it to a target location. The agent must manipulate each finger and wrist joint with dexterity to grasp the object correctly and learn the correct ball and target positions. An observation consists of hand joint angles, the object position/orientation, and the target position/orientation. We

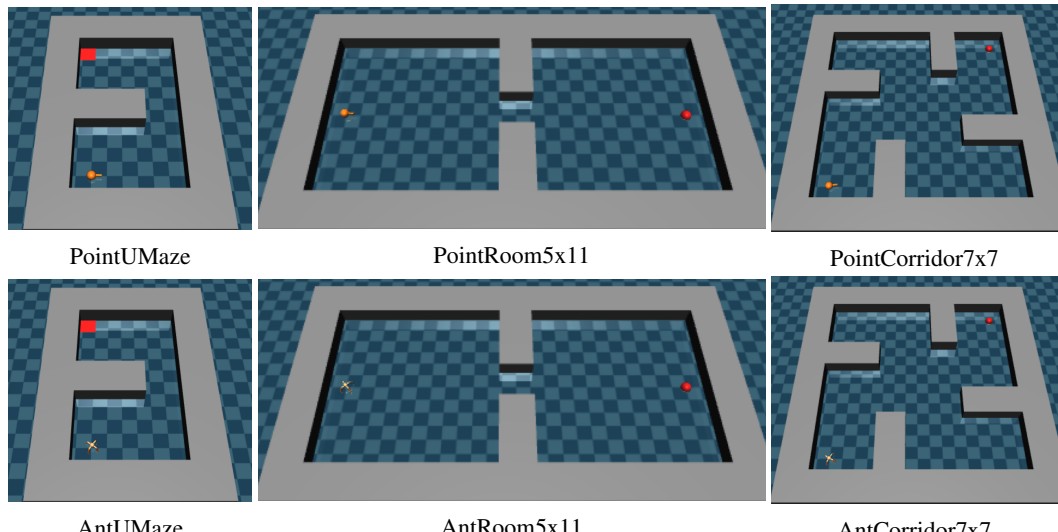

PointUMaze          PointRoom5x11          PointCorridor7x7

AntUMaze          AntRoom5x11          AntCorridor7x7

Figure 8: Maze environments. There are two agents: Point (top row) and Ant (bottom row). The agent must learn to move to the goal (red). Only the goal has a positive reward, all other states have a reward of zero.

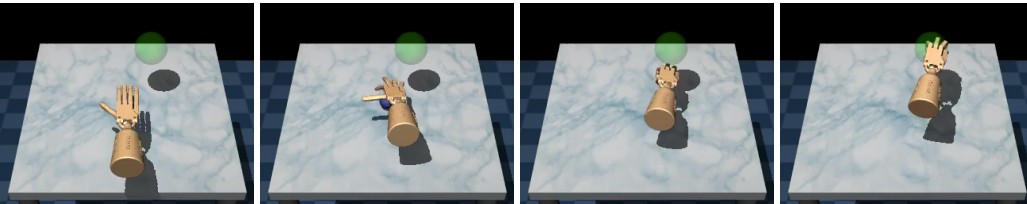

Figure 9: Adroit Relocate environment. The goal is to pick up the ball and move it to the target (green).

modify the original domain by fixing the initial qpos of the hand and by terminating the episode when the agent correctly solves the task.

# B   Experiment Details

## B.1   Expert demonstrations

For the Fetch environments, we use pre-defined settings in [28] to train an expert policy. For the maze environments, we predefine a series of subgoals, and the position of the next subgoal is concatenated to the state. The reward is dense, using the negative Euclidean distance to the next subgoal and subsequent subgoals, and we use soft actor-critic [8] as the expert policy. For the Adroit environment, we use the pre-trained policy from [30].

On the Fetch domains, as each episode is of length 50, this amounts to a total of a couple of hundred training samples. We thus make 10 identical copies of each demonstration in order to stably train the policy and backwards model (we do the same for all other baselines). This has a similar effect of performing more gradient updates, but we found it to be computationally cheaper. For both Maze and Adroit environments, we use 20 demonstrations and do not repeat the demonstrations as the episodes are sufficiently long.

## B.2   Hyperparameters

All hyperparameter settings for BMIL are provided in Tables 3 and 4. For all experiments, we used an internal cluster with single GPU compute nodes, with 10 virtual CPU cores and a GPU with either an NVIDIA P100 or V100.

For all methods, we use the same number of policy gradient steps for a fair comparison. For the VINS baseline, we rely on the hyperparameter settings provided in the paper for the Fetch environments and also run hyperparameter sweeps to find the best settings specific to our environments. We run longer hyperparameter sweeps for the Maze and Adroit environments.

| | | Fetch | Adroit |
|---|---|---|---|
| | Push | PickAndPlace | Relocate |
| epochs | | 200 | 600 |
| policy updates per epoch | | 100 | 50 |
| batch size | | 64 | |
| demonstrations | 5 | 10 | 20 |
| demonstration sampling ratio $p$ | | 0.5 | 0.8 |
| trace horizon length | 1 | $1 \to 3$ $(1 \to 200)$ | $1 \to 10$ $(100 \to 600)$ |
| action selection strategy | | entropy | |
| action selection coefficient | | 30 | 3 |

Table 3: Hyperparameters for Fetch and Adroit. $x \to y(a \to b)$ denotes a thresholded linear function as used in [12], implemented as $f(e) = \min\left(\max\left(x + \frac{e-a}{b-a}(y-x), x\right), y\right)$ for epoch $e$.

| | Point | | | Ant | | |
|---|---|---|---|---|---|---|
| | UMaze | Room5x11 | Corridor7x7 | UMaze | Room5x11 | Corridor7x7 |
| epochs | 800 | | | 400 | | |
| policy updates per epoch | 250 | | | 500 | | |
| batch size | 256 | | | | | |
| demonstrations | 20 | | | | | |
| demonstration sampling ratio $p$ | 0.8 | | | 0.95 | 0.9 | 0.95 |
| trace horizon length | $1 \rightarrow 10$ $(100 \rightarrow 800)$ | | | $1 \rightarrow 10$ $(100 \rightarrow 400)$ | | |
| action selection strategy | entropy | | | | | |
| action selection coefficient | 40 | 1 | | 40 | 10 | |

Table 4: Hyperparameters for Maze. $x \rightarrow y(a \rightarrow b)$ denotes a thresholded linear function.

# C   Results

## C.1   Main results

| | | | BC | VINS | BMIL |
|---|---|---|---|---|---|
| FETCH | Push (5 demos) | | $99.8_{\pm0.5}$ | $98.1_{\pm0.6}$ | $99.5_{\pm0.7}$ |
| | PickAndPlace (10 demos) | | $100_{\pm0.0}$ | $98.8_{\pm0.8}$ | $99.2_{\pm0.8}$ |
| MAZE | Point (20 demos) | UMaze | $95.7_{\pm1.3}$ | $7.1_{\pm2.3}$ | $71.6_{\pm8.0}$ |
| | | Room5x11 | $96.0_{\pm1.5}$ | $26.9_{\pm6.2}$ | $95.7_{\pm2.6}$ |
| | | Corridor7x7 | $87.2_{\pm1.7}$ | $66.8_{\pm4.0}$ | $90.6_{\pm2.8}$ |
| | Ant (20 demos) | UMaze | $100_{\pm0.0}$ | $39.9_{\pm5.6}$ | $100_{\pm0.1}$ |
| | | Room5x11 | $97.3_{\pm0.8}$ | $93.5_{\pm1.3}$ | $96.4_{\pm0.8}$ |
| | | Corridor7x7 | $95.9_{\pm0.9}$ | $88.5_{\pm1.7}$ | $90.1_{\pm1.6}$ |
| ADROIT | Relocate (20 demos) | | $100_{\pm0.0}$ | $96.0_{\pm1.7}$ | $100_{\pm0.0}$ |

Table 5: Success rates during training for Fetch, Maze, and Adroit environments over $400,100,100$ trials, respectively. The bounds indicate 95% confidence intervals. BC and BMIL consistently solve the original task with the initial start and goal states for all environments, while the success rates for VINS fluctuate in the Maze environments.

Table 5 shows the success rates during training, on environments where the start and goal positions are unchanged. We see that both BC and BMIL achieve high success rates across all environments, while VINS does not for the Maze environments.

## C.2   Additional Experiments

For all additional experiments on the Fetch domains, we use 100 trials for each method. The error bars or shaded regions denote 95% confidence intervals.

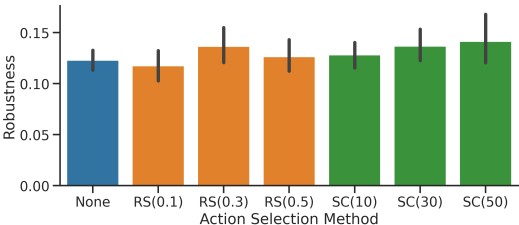

Figure 10: Action selection strategy on Push. We see a similar trend as in PickAndPlace, in that perturbing the action increases robustness over None.

Figure 10 shows the robustness of different action selection strategies on Fetch-Push. Here we see that all action selection strategies generally perform similarly, with possibly RS(0.3) and SC(50) a slight edge over the no strategy (None), though the error bars are fairly large.

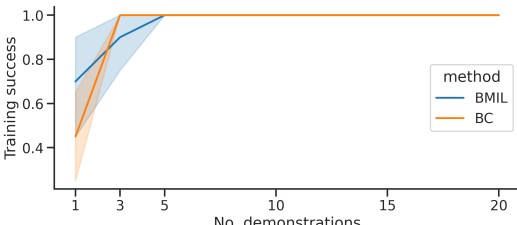

Figure 11: Number of demonstrations: training success rates on PickAndPlace.

Figure 11 shows the success rates during training with a varying number of demonstrations on Fetch-PickAndPlace. We see that 3 demonstrations are sufficient for BC to achieve a $100\%$ success rate on the original start and goal positions. However, BMIL requires at least 5 demonstrations to attain the same level of performance, as the backwards model requires at least a certain number of samples to train in a stable manner. We use 10 demonstrations in our experiments.

| | Robustness (%) | | | Relative to BC | | |
|---|---|---|---|---|---|---|
| | BC | BMIL | BMIL (model first) | BC | BMIL | BMIL (model first) |
| Push (5 demos) | $12.1_{\pm0.3}$ | $14.6_{\pm0.6}$ | $13.2_{\pm1.0}$ | 1 | 1.21 | 1.09 |
| PickAndPlace (10 demos) | $4.1_{\pm0.1}$ | $17.5_{\pm0.9}$ | $19.9_{\pm2.4}$ | 1 | 4.31 | 4.85 |

Table 6: Training the backwards model first and then the policy produces similar results as training them together (BMIL).

Table 6 shows the results of training the backwards model first compared to BMIL which trains both the model and the policy in the same training loop. We find similar robustness values, where training the model first produces slightly lower values on Push and slightly higher values on PickAndPlace.