# OpenReview forum: "Robust Imitation of a Few Demonstrations with a Backwards Model"
_NeurIPS.cc/2022/Conference — NeurIPS 2022 Accept_

### Official Review · Reviewer_VFop · 2022-07-10

**Rating:** 6
**Confidence:** 5
**Soundness:** 3 good
**Presentation:** 4 excellent
**Contribution:** 2 fair

**Summary:**

This paper seeks to improve the robustness of traditional behavior cloning, by aggregating backward traces from the demonstration to nearby states into the dataset. They do so by learning a factored backwards dynamics model, and rolling out this dynamics model from demonstration states to retrieve backward traces. By training a subsequent learner policy on original demonstrations and backward traces that lead the agent back to the demonstrated states, they improve the robustness of the learned policy on 4 different domains.


**Questions:**

In addition to the primary in-line questions mentioned in the above section, here are a few other questions I have -
1. In Line 148-149, the paper mentions irrecoverable states. The reviewer agrees that the assumption of existence of recovery actions is reasonable outside of such irrecoverable states. However, does this preclude the use of the approach in domains where irrecoverable states exist? For example, could a backwards model be queried at states nearby irrecoverable ones, and be made to be more robust near such states?
2. How do the authors balance the backwards model B being accurate w.r.t the forward policy, with its ability to generate diverse new traces? (as mentioned in lines 163-164).
3. How reasonable are the assumptions of Lines 179-180, of “”If the reverse time model B is accurate and the previous action generator BA gives sufficiently diverse actions”?
4. In Line 195, is the definition of $a’$ meant to be $a’ \sim \mathcal{N} ( \mu, \sigma), \mu = [ a - k, a + k]$, rather than the specified equation?
5. In line 191, why is it necessary for the action-selection strategy to cover the expert actions? Is this to maximize robustness? What happens to robustness if the action selection distribution and the expert actions only intersect?
6. In Line 198, the paper states “As we do not wish to deviate too much from the demonstration data, we perturb only the first action when $s’ \in \mathcal{D}_{demo}$”. Why is this desirable? Why don’t we want to stray far away from the demonstration data?
7. In Lines 203-205, the paper states “In addition to using relatively short rollouts that limits compounding errors, we hypothesize that this is these traces do not necessarily need to be accurate in order to benefit the policy. It may be that having the general correct direction contained in the traces is sufficient for the policy to eventually reach states in the demonstration data”. By this logic, isn’t it sufficient to have an arbitrary action selection strategy, rather than one that relies on the backwards dynamics?
8. How are the forward dynamics models trained in Line 298 used?


**Limitations:**

For limitations, please view the section on weaknesses above.


**Strengths And Weaknesses:**

### Strengths

** Clarity: **

1. Overall, the paper is written very well. The problem tackled is well motivated, the approach is described in detail and most design choices are well justified. The contribution is well contextualized in the context of prior work, and the experimental results are also described in sufficient detail. Two particular ways in which the paper is written that should be appreciated are -

    a. The paper does not exaggerate their claims beyond what their experimental results show.

    b. The work also pre-emptively answers several questions that naturally arise during the discussion of their work (such as the accuracy of the backward dynamics model, and the effect it has on robustness of the learnt policy).

** Quality: **

2. Overall, the paper is also technically sound.

    a. The paper honestly describes their claims (more robustly solving a set of tasks that current IL algorithms can currently solve), and their claims are supported by the experimental results they present.

    b. The choices of methods are well justified, and for the most part, the paper provides intuition about the functioning of their method (such as the dynamics of accuracy of the backwards model in training, and its relation to how robustly the policy learns).

    c. I also appreciate that the paper is very upfront about which specific problem setting they tackle, and when their approach would, and would not, be useful.

### Neutral

** Originality **
1. The originality of the paper is fair. The paper adopts a very similar approach to prior work [41], but applied to the imitation learning domain rather than the reinforcement learning domain as in [41]. This suggests the approach itself is not entirely novel, and only differs in some of the lower-level implementation details. This; however, is acceptable - the paper is upfront about this similarity to prior work, but clearly discusses why solving a similar problem in the imitation learning domain is important, why current methods that do so are limited (such as DAgger being expensive to run, etc.). As a result, one must give credit to the paper for identifying the applicability of this approach to the imitation learning setting. As a result, the overall originality of the paper is fair.

### Weaknesses

** Significance **
1. While the experimental results presented are technically sound, they are presented in somewhat limited settings.

    a. The tasks that the paper evaluates their approach on are quite simplistic tasks, such as solving a maze or controlling the Fetch end-effector to reach / push / place objects. In these tasks, “robustness” of the policy is a nice property, but not crucial to success or necessary for safety. The paper could benefit from evaluating their approach against baseline approaches on a more diverse set of tasks that are inherently harder to solve. Examples of such tasks could be more complex manipulation tasks, tasks in the joint space, continuous navigation tasks, and more interestingly, tasks where dynamics matter as well, such as continuous control of cars or legged agents. While I appreciate the authors mention this is a plan in future work, the impact of the paper in its current form is a tad underwhelming, and could be greatly improved by such evaluation on diverse tasks.

2. Applicability of approach. The paper could also benefit from both discussing, and experimenting on, a broader settings (in addition to other tasks as mentioned above). The paper currently states such settings are not considered, but it is unclear why. Some such settings include -

    a. For some reason, the paper only explores robustness with respect to spatial dimensions (such as joint coordinates, and not velocities, as mentioned in Lines 143-144). It is unclear why they restrict their discussion to this setting. Being able to recover to arbitrary states, and not just spatial dimensions, would be a big improvement to the paper. Also, several interesting questions then arise - does one need to follow a different action selection strategy for such dimensions? Does one need to train the backwards dynamics model any differently for such cases? Are backwards dynamics models as well conditioned when joint velocities or other non-linear dynamics are concerned?

    b. Another interesting setting is when there are additional demonstrations available. Quantifying how much the robustness improves from the proposed approach with respect to number of additional demonstrations / some measure of coverage of state / action space of the demonstrations would be interesting to see, and useful in assessing the impact of the paper.

    c. Another interesting area (but one that is likely better suited for future work) is when there is also access to computationally expensive interactive experts and demonstrations. Could the proposed approach be combined with interactive feedback to verify or improve robustness of the learner?

### Note to authors
I strongly urge the authors to consider evaluating their approach on the above suggested domains / settings, as I believe it would contribute to making their paper much stronger. Currently, my biggest reservation with the paper is these evaluations, but should these be improved, I think this could make for a strong paper.

---

> ### Author Response · Authors · 2022-08-02
> **Response to specific questions and concerns [2/2]**
>
> > How do the authors balance the backwards model B being accurate w.r.t the forward policy, with its ability to generate diverse new traces?
>
> We argue that generating more diverse traces does not directly hinder the backward model's ability to learn accurately w.r.t the forward policy. The backwards model was trained with maximum likelihood on the demonstrations so we wish the model to be as accurate as possible on this data. When generating traces, we wish to generate new plausible state-action pairs but are limited by the backwards model's extrapolation capabilities. We therefore try not to deviate too much from the support of the demonstrations (where errors would become larger) and only perturb the first action. Furthermore, the traces are only used to train the policy and not the backwards model, so perturbing traces does not affect the model's accuracy on the demonstration data.
>
> > How reasonable are the assumptions of Lines 179-180, of “If the reverse time model B is accurate and the previous action generator BA gives sufficiently diverse actions”?
>
> There is some evidence that neural networks can extrapolate somewhat on points sufficiently close to the training data (Xu et al. 2021, How Neural Networks Extrapolate: From Feedforward to Graph Neural Networks). As we use short horizons for generating traces and we only perturb the first action by scaling the variance, we believe that the backwards model is reasonably accurate along the traces and does not give infeasible state-action pairs, although there is some extrapolation error. Empirically, we find that these traces still improve robustness over BC across a variety of domains and types of tasks. We have included this discussion and reference in Section 3.3.
>
> > In Line 195, is the definition of $a'$...rather than the specified equation.
>
> Thank you for pointing this out. The noisy action $a'$ is found by simply adding random uniform noise to the original action $a$. We will simplify the equations to be $a' = a + k, a \sim \mathcal{N}(\mu, \sigma^2), k \sim \mathcal{U}[-k, k]$.
>
> > In line 191, why is it necessary for the action-selection strategy to cover the expert actions?
>
> We apologize for the unclear wording. It is not required for the action selection strategy to cover the expert actions but it is a consequence of trying not to stray too far from the demonstration data. Intuitively, we wish to generate diverse traces that are plausible and do not contain state-action pairs that are incorrect. However, the further we move away from the demonstration data, the more inaccurate the model becomes. Thus actions that are closer to the expert actions should be more likely, resulting in a bell-curve shape centered at the expert actions. We have significantly revised the action selection strategy discussion for clarity.
>
> > Why is this desirable? Why don’t we want to stray far away from the demonstration data?
>
> This is because we want to limit model misprediction errors and be able to generate diverse traces that are reasonable and feasible. We are balancing the tradeoff between model error and generating unseen traces.
>
> > By this logic, isn’t it sufficient to have an arbitrary action selection strategy, rather than one that relies on the backwards dynamics?
>
> This would be true if we had an oracle backwards dynamics model so we could sample arbitrary actions with diverse directions to further robustify the policy. However, because the backwards model is only trained on the demonstration data, we need to limit its misprediction accuracy. In our experiments, perturbing the traces slightly (where there is clearly non-zero model error) lead to a more robust policy (Section 5.2 action selection strategy). As such, we hypothesized that having slightly incorrect state-action pairs are still beneficial for robustness as the policy may only rely on the general direction of the trace. For example, in a maze with few obstacles as in our domains, an agent need not follow the trace exactly in order to reach the goal. It can get away with landing slightly ahead or behind at different timesteps, or can even be slightly off the trace. However, following the general direction of the trace would help the agent get closer to the goal where there's a higher chance of success.
>
> > How are the forward dynamics models trained in Line 298 used?
>
> The forward dynamics model is used similarly to the backwards dynamics model and generates new traces for training the policy. From every $(s, a)$ in the demonstration data, the forward dynamics model predicts the next state, and the current policy is used to predict the next action. This process is rolled out to the specified horizon length. This process is very similar to Dyna-style model-based RL algorithms (e.g., MBPO from Janner et al. 2019), with the exception of no environment interactions. We have added these details in the draft.

---

> > ### Comment · Reviewer_VFop · 2022-08-07
> > **Response to Author' Rebuttal**
> >
> > I thank the authors for their response and clarifications. Some of the questions I had / clarifications I required have been made, such as the intuition of staying close to the expert demonstrations. I request the authors to ensure that appropriate clarifications have been also made to the text of the paper.
> >
> > That said, I still have few follow-up concerns, primarily regarding the evaluations of the approach. Particularly -
> >     1. I appreciate the evaluation of the approach on a new environment. This experiment is a welcome addition to the paper. However, this new environment still suffers from similar characteristics to the fetch environments, namely that in such tasks, the “robustness” of the policy is a nice property, but not crucial to success or necessary for safety.
> >     2. There is still no exploration of perturbations of non-spatial dimensions. Even in the new environment, the newly perturbed dimensions are still spatial dimensions / initial positions, as opposed to velocities or other types of states. The paper would greatly benefit from being able to show robustness to other arbitrary dimensions.
> >     3. The specified equation for a' suggests k is sampled from $\mathcal{U}[-k,k]$ - this notation is unclear, can the authors clarify?
> >
> > Considering the evaluation on the new domains, but also these concerns I update my rating of the paper to 6.

---

> > > ### Author Response · Authors · 2022-08-08
> > > **Further response to feedback**
> > >
> > > We thank the reviewer for taking the time to respond to our rebuttal and for raising their score. We will update our paper accordingly and add more detailed explanations (perhaps with an example) of the intuition behind our method. We hope we have addressed all remaining concerns below.
> > >
> > > > However, this new environment still suffers from similar characteristics to the fetch environments, namely that in such tasks, the “robustness” of the policy is a nice property, but not crucial to the success or necessary for safety.
> > >
> > > That's a valid point. All of the domains considered do not require robustness for success or safety, but robustness is still desirable as it allows using fewer demonstrations and the demonstrations can have smaller coverage of the state-action space. We will try to extend this method to domains where robustness is needed, and possibly consider the case of irrecoverable states in future work.
> > >
> > > > Even in the new environment, the newly perturbed dimensions are still spatial dimensions / initial positions, as opposed to velocities or other types of states.
> > >
> > > For Adroit in particular, the joint velocities are not included in the observation, so the agent sees only the joint positions of the hand. The other remaining dimensions are miscellaneous information required to solve the task such as the object or target position. We chose not to perturb these dimensions as this would now become more of a multi-goal setting, which is orthogonal to our robustness setting. We will try to perform additional experiments in the revision where we perturb joint velocities in the Fetch/Maze environments. We could also look into extracting the joint velocities in Adroit to include it as part of the state and perturb these dimensions as well.
> > >
> > > > The specified equation for a' suggests k is sampled from $\mathcal{U}[-k, k]$ - this notation is unclear, can the authors clarify?
> > >
> > > Thank you for pointing this out; we accidentally reused $k$ as both a sample and a parameter of the distribution when updating the equation. The notation should be $a' = a + k, a \sim \mathcal{N}(\mu, \sigma^2), k \sim \text{Unif}[-b, b]$, where $b$ is a hyperparameter. This will be updated in the final draft.

---

> ### Author Response · Authors · 2022-08-02
> **Response to specific questions and concerns [1/2]**
>
> > The tasks that the paper evaluates their approach on are quite simplistic
>
> In the maze/2D nav environments, the agent observes joint positions/velocities and its current position and must control each joint to reach the maze goal. This requires learning continuous locomotion over a long horizon and thus the maze environments are quite difficult. In particular, the ant mazes used in our experiments are examples of continuous control of legged agents. We additionally perform experiments on a new dexterous manipulation domain (Rajeswaran et al., 2018). The task is to manipulate an Adroit hand to pick up a ball from the table and move it to a target location. We fix the starting positions of the hand, ball, and target and evaluate the robustness by adding uniform noise to the original hand joint positions. Over 100 trials, we see that BMIL improves robustness over BC by $68\%$ on this complex task. These results have been added to Table 1 and we will add more Adroit tasks in the camera-ready.
>
> |  | Robustness (%) |  | Relative to BC |  |
> |---|---|---|---|---|
> |  | BC | BMIL | BC | BMIL |
> | Adroit Relocate (20 demos) | $7.9 {\scriptstyle \pm 0.7}$  | $13.3 {\scriptstyle \pm 1.0}$ | $1.00$ | $1.68$ |
>
> > the paper only explores robustness with respect to spatial dimensions...does one need to follow a different action selection strategy for such dimensions?
>
> Good point. First, during trace generation, we perturb only the action and thus no knowledge of the individual dimensions of the observation space is required. If one knows how each action dimension affects each observation dimension, then utilizing a finer-grained action selection strategy (e.g. use different scaling coefficients for each action dimension) would produce better results. During evaluation, we varied the spatial dimensions as we were primarily concerned with learning the region of attraction around the demonstrated path, but we agree that recovering from arbitrary states is useful to know. In the new Adroit domain, we vary not just the initial position but the qpos of the hand (30-dim) by adding random uniform noise, and find that BMIL can recover from arbitrary states and not just positions.
>
> > Another interesting setting is when there are additional demonstrations available.
>
> While we consider the setting where demonstrations are costly to obtain, we also perform experiments where additional demonstrations are available in Section 5.2 (Figures 4b and 9). With more demonstrations, we find that both BC and BMIL learn more robust policies, but up to a certain threshold where robustness plateaus. BC plateaus at a much lower number of demonstrations than BMIL.
>
> > some measure of coverage of state/action space
>
> As we fix the initial state and goals in our domains, the expert trajectories naturally have a very narrow coverage of the state-action space. It would be interesting to see how the improvement in robustness changes as the demonstrations have increasing state-action space coverage. We surmise that having wide-covering demonstrations would likely increase the base success rates of both BC and BMIL by about the same amount, but BMIL's improved robustness from diverse traces would still hold.
>
> > Another interesting area (but one that is likely better suited for future work) is when there is also access to computationally expensive interactive experts and demonstrations.
>
> In our problem setting of an offline expert with only state-action pair demonstrations, we had to use a relatively computationally expensive model on top of BC to improve robustness in this highly restrictive setting. If interactive feedback is available, we think that an approach such as DART (Laskey et al. 2017), which adds noise to demonstrations and labels new observations with the expert, would be more helpful. An interesting extension would be to possibly extend DART with a backwards dynamics model.
>
> > However, does this preclude the use of the approach in domains where irrecoverable states exist?
>
> Interesting question. We assumed no irrecoverable states as the backwards model should not generate reverse traces that are simply not feasible. However, if there is prior knowledge of the set of irrecoverable states $\mathcal{S}_I$, then we could simply resample $s, a \sim p(s, a \mid s')$ during trace generation if $s \in \mathcal{S}_I$ and only generate traces that do not contain irrecoverable states. We could further avoid generating states which are too close to irrecoverable states so that the policy learns to avoid unsafe regions within the funnel. This would be a good avenue to explore in future work.

---

### Official Review · Reviewer_rhnT · 2022-07-11

**Rating:** 6
**Confidence:** 4
**Soundness:** 3 good
**Presentation:** 4 excellent
**Contribution:** 3 good

**Summary:**

This work introduces a novel method that incorporates a backward model into the offline imitation learning process to improve the model robustness against covariate shifts. The generated demonstrations from the backward model would train the policy to recover from the error states and back to the main distribution that it is familiar with. The motivation is clear and the idea of using offline data augmentation is a promising direction for improving the robustness of imitation learning algorithms.

**Questions:**

1. Why train the backward model together with the policy model? As an offline method, I thought the easiest way is to first train the backward model alone and then train the policy alone. I hope the authors could give more details and motivation about the training pipeline.
2. What’s the observation used in the Fetch experiment? I assume it is the proprioception / joint angles of the Fetch arm. Do you give the target pose of the object? Is the proposed method limited to low-dimensional observations? Any discussion about potential high-dim observation inputs like images would be great.
3. How does the method perform if the given demonstration is sub-optimal? Let’s say the demonstration is collected by a human expert and there are imperfect motions. Will it be a challenge for this method?

**Limitations:**

1. Bi-directional state transition assumption.
2. Low-dimensional observation space.
3. Might be hard to handle sub-optimal demonstrations.

**Strengths And Weaknesses:**

Strength:
1. The paper is well-written. The problem setting and technical terms are clearly introduced.
2. The idea of learning a backward model to generate more offline demonstrations is novel. Most prior imitation learning works only focus on the method part, I believe a good way of augmenting the training data is also an important direction to improve the robustness of offline imitation learning.
3. The idea of decomposing the backward model into two functions is a good way to model the different features of the action generation and state transitions. To handle the multi-model distribution of action generation, a conditional Gaussian probabilistic model is used, which could further generate diverse trajectories by sampling the noise. I found these architectural designs promising and reasonable.
4. The experiment results in both robot and maze environments show the promising advantages of the proposed approach.

Concerns:
1. One assumption is missing: this method assumes that the state transition in the environment is bidirectional. While in many real-world scenarios, the forwarding trajectory is not reversible which might make the generated traces impracticable in the actual environment. The proposed method might struggle when the environment has many bottleneck state transitions that are not reversible. But I do agree this is out of the main focus of this work.
2. It seems like the experiment tasks are more like some proof of concept. I'm a little bit concerned about the performance in high-dimensional and complex tasks. For example, what if the observation is an image?
3. I do agree generating more trajectories from given demonstrations could improve the robustness of the learned policy. But it is still unclear to me why the model could generate novel initial states and train the policy to recover from it. In the Fetch experiments, it seems like  BMIL solved many cases with unseen initial states. But these states shouldn’t be generated by the backward model since it is unseen in the given demonstrations. Then, why the policy model could handle it so well?

---

> ### Author Response · Authors · 2022-08-02
> **Response to specific questions and concerns**
>
> > this method assumes that the state transition in the environment is bidirectional.
>
> Thank you for pointing this out. We alluded to this in the paper but failed to state this exactly. We have added a sentence to make this assumption explicit in Section 3.2.
>
> > I'm a little bit concerned about the performance in high-dimensional and complex tasks. For example, what if the observation is an image?
>
> Good point. Our work does not specifically preclude image observations as we only assume a notion of similarity for the outcome of actions: perturbing an action with a small amount of noise will lead to new next states close to the original next state. However, BMIL will likely not work for image inputs in its current form. To handle high-dimensional states, more powerful model architectures and possibly an encoder to learn good latent representations would probably be required. A finer-grained action selection strategy may also be needed to produce diverse next states. We plan to extend to image inputs in future work.
>
> Regarding more complex tasks, we have added a new challenging dexterous manipulation domain (Rajeswaran et al., 2018) where the goal is to control the joints of a 24-DoF Adroit hand to pick up a ball and move it to a target location. We fix the initial states during training and vary the initial hand joint positions during evaluation. In this difficult domain, our method produces a policy that is much more robust than BC. We have added this result to the paper and will perform experiments on another Adroit task before publication.
>
> |  | Robustness (%) |  | Relative to BC |  |
> |---|---|---|---|---|
> |  | BC | BMIL | BC | BMIL |
> | Adroit Relocate (20 demos) | $7.9 {\scriptstyle \pm 0.7}$  | $13.3 {\scriptstyle \pm 1.0}$ | $1.00$ | $1.68$ |
>
> > But these states shouldn’t be generated by the backward model since it is unseen in the given demonstrations.
>
> Great question. Regarding novel states, as we perturb the action, the state-action pair that is input to the backwards model is now out of the support of the training data to the model. Since this process is rolled out, the output would also be a novel state-action pair. We will verify that the backwards model generates new states with a small experiment before publication. Now whether this unseen state-action pair is actually feasible depends on the model's generalization or extrapolation capabilities. There is some evidence that neural networks exhibit graceful deterioration during extrapolation the further away we move from the training data (Xu et al. 2021, How Neural Networks Extrapolate: From Feedforward to Graph Neural Networks). To limit these errors so that the model can generate unseen but realistic states, we perturb only the first action slightly and limit the trace length to a small number. Extrapolation is also likely made easier by the fact that we use low-dimensional states. We have added this explanation more clearly in Section 3.3 action selection strategy.
>
> > Why train the backward model together with the policy model?
>
> We trained the model with the policy in a single loop as this was slightly faster in computation time. However, we can indeed train the backwards model first and then train the policy separately. Using this method, we perform 100 trials on the Fetch tasks and find similar results as our original algorithm. On `Push`, our original algorithm performs slightly better while training the model first gives slightly better results on `Pick`.
>
> |  | Robustness (%) |  |  |
> |---|---|---|---|
> |  | BC | BMIL | BMIL (Train model first) |
> | Push (5 demos) | $$12.1 {\scriptstyle \pm 0.3}$$ | $$14.6 {\scriptstyle \pm 0.6}$$  | $$13.2 {\scriptstyle \pm 1.0}$$ |
> | Pick (10 demos) | $$4.1 {\scriptstyle \pm 0.1}$$ | $$17.5 {\scriptstyle \pm 0.9}$$ | $$19.9 {\scriptstyle \pm 2.4}$$ |
>
> > the observation used in the Fetch experiment?
>
> The observation space is 25-dimensional and consists of the end effector coordinates and its linear velocity, the gripper's position and velocity, and the object's pose, velocities, and its relative position/velocity to the gripper. We have added descriptions of observations for all domains in the revision.
>
> > How does the method perform if the given demonstration is sub-optimal?
>
> We assume that the expert trajectories consist of only state-action pairs without rewards. In this limited setting, one cannot discern if the given trajectory is optimal, so our interpretation of optimality here is how closely we can follow demonstrations. This is in fact a feature of our method as we do not have to modify rewards or values to be conservative as done in many offline RL papers (e.g. Fujimoto et al. 2019, Kumar et al. 2020). If the expert trajectories were sub-optimal to begin with, then BMIL would simply follow the demonstrated path and learn regions of attractions around that path as intended. In this case, we would need a different evaluation metric to measure robustness instead of task completion success rates.

---

> > ### Comment · Reviewer_rhnT · 2022-08-09
> > **Thanks for your clarification**
> >
> > I thank the authors' reply. Some of my concerns are addressed for example the details of training the backward model together with the policy and the assumption of the proposed method. However, I still feel the task setup is limited to low-dimensional inputs. The added dexterous manipulation is similar to the fetch task regarding the environment complexity. Therefore, I will maintain my original score 6.

---

### Official Review · Reviewer_L7Ps · 2022-07-12

**Rating:** 5
**Confidence:** 4
**Soundness:** 3 good
**Presentation:** 3 good
**Contribution:** 3 good

**Summary:**

The paper presents a simple method to robustify policies that are trained from a small number of demonstrations using supervised learning (i.e., behavior cloning). The key idea is to synthesize recovery trajectories that are not included in the training set. This is done by generating possible short-horizon trajectories that end at the demonstrated state using a learned backward dynamics model $p(s_t, a_t| s_{t+1})$. The trajectories are then used as additional data to train the behavior cloning policy. The backward dynamics models are encouraged to generate diverse trajectories in order to enrich the set of out-of-distribution states that the policy can handle. Both the policy and the backward dynamics model are trained from a small number of demonstrations. The method is shown to achieve better robustness compared to naive behavior cloning method and a competitive offline IL baseline (VINS) on two simulated environments.

**Questions:**

Please refer to the main comments listed above.

**Limitations:**

The paper does not mention any limitation of the proposed method. Like mentioned in the comments above, I hope the authors could stress test the method and discuss the limitations of the method both conceptually and empirically.


**Strengths And Weaknesses:**

Overall I enjoyed reading the paper. The method is conceptually simple and easy to apply on top of existing offline imitation learning methods. The empirical results also seem to support the main claims of the paper.

My main concern about the paper is the scope and the details of the experimental evaluation.

- The evaluation tasks are rather simple both in terms of the nature of the tasks (reaching / pushing & 2D nav) and the types of dynamics involved. Hence it is unclear to me whether the method is still effective for more complex tasks such as Franka Kitchen and dexterous manipulation (Fu et al., 2020). Given the ease of implementation and the fact that these environments have expert data available, I’d strongly encourage the authors to stress test the proposed method on these harder tasks.
- How is the forward dynamics model baseline implemented? Specifically, how does the policy make use of the rollout generated by the forward dynamics model?
- How does the accuracy of the backward dynamics model influence the final robustness performance? I’d suggest a simple ablation study that injects different levels of noise to the learned backward dynamics model.
- According to Table C.1, even BC can achieve near perfect performance in both Fetch and Maze tasks with only 5-20 demonstrations. Unless I’m missing something, this seem to directly contradict with the results in prior works (e.g., in D4RL) that require a few orders of magnitude more data to achieve reasonable performance. I hope the authors can explain this discrepancy or point out my misunderstanding.

---

> ### Author Response · Authors · 2022-08-02
> **Response to specific questions and concerns**
>
> > The evaluation tasks are rather simple both in terms of the nature of the tasks and the types of dynamics involved.
>
> We agree that more complex tasks are useful for evaluation, though we argue that the maze/2D nav environments are quite challenging as they require continuous locomotion over a long horizon (states are the agent's joint positions/velocities, current timestep, and the agent's current position). We also performed additional experiments on the suggested dexterous manipulation domain (Rajeswaran et al., 2018, Fu et al., 2020) on the `Relocate` task where the goal is to pick up the ball with the Adroit hand and move it to a target location. We fix the starting hand, object, and target positions and evaluate policy robustness by adding uniform noise to the original hand joint positions. We use 20 demonstrations and run for 100 trials. We see that BMIL significantly improves robustness over BC by roughly $68\%$. We have added these results to Table 1 and will add another Adroit task for the camera-ready.
>
> |  | Robustness (%) |  | Relative to BC |  |
> |---|---|---|---|---|
> |  | BC | BMIL | BC | BMIL |
> | Adroit Relocate (20 demos) | $7.9 {\scriptstyle \pm 0.7}$  | $13.3 {\scriptstyle \pm 1.0}$ | $1.00$ | $1.68$ |
>
> > Specifically, how does the policy make use of the rollout generated by the forward dynamics model?
>
> Starting from every state-action pair within the demonstrations, the forward dynamics model predicts the next state and the current policy predicts the next action. The trace is rolled out from this next state-action pair until the horizon length is reached. As with the backwards model, the policy is trained on both the demonstrations and these forward rollouts. This process is nearly identical to Dyna-style model-based RL algorithms (e.g., MBPO from Janner et al. 2019). We have added more explanation to the forward dynamics model experiments in Section 5.2.
>
> > How does the accuracy of the backward dynamics model influence the final robustness performance? I’d suggest a simple ablation study that injects different levels of noise to the learned backward dynamics model.
>
> This is indeed important to investigate and we try different action selection strategies and noise levels in Section 5.2 "Action selection strategy". We find that while the optimal level of noise is different for each domain, adding some noise using the variance scaling (SC) strategy leads to more robust policies than no noise.
>
> > According to Table C.1, even BC can achieve near perfect performance in both Fetch and Maze tasks with only 5-20 demonstrations.
>
> Prior works (e.g. D4RL) use demonstrations for random object and goal locations. In our work, as our objective is to measure robustness to deviations from the demonstrated path, we fix the object/goal locations and vary the initial state during evaluation. Our setting covers a much narrower subset of the state-action space, resulting in much fewer demonstrations required overall. We have included a sentence to clarify this discrepancy in Section 3.2.
>
> > The paper does not mention any limitation of the proposed method.
>
> The limitations of our method are mainly due to the restrictive problem setting as outlined in Section 3.2, with no access to experts or the environment. We further restrict ourselves to state-action only demonstrations. The environment must not contain any irrecoverable states so that the backwards dynamics model does not generate any unsafe traces (the policy should avoid such paths/regions). We also use low-dimensional states instead of image inputs and mention that BMIL requires around 5x more computation than BC on the Fetch domains (Section 5.2 Computation budget). Future work will include how to extend BMIL to accommodate for less restrictive settings such as one with environment interactions, online experts, or high-dimensional image inputs.

---

### Author Response · Authors · 2022-08-02
**Summary of Response and Revisions**

We thank the reviewers for the detailed feedback and helpful suggestions and hope that we have addressed all concerns. The reviewers all appreciate the concept of pairing a backwards dynamics model with offline imitation learning to improve policy robustness and the presentation of the method. The reviewers are concerned that the evaluation tasks are relatively simple and limited. We clarify the difficulty of the maze environments, especially with the ant robot, and also include a new Adroit dexterous manipulation environment. In this challenging domain, BMIL still improves robustness by approximately $68\%$ over BC when evaluated on random initial states. We plan to include another Adroit task before publication.

We have also uploaded a revised draft (changes are in blue for the reviewer's convenience). Summary of revisions:

1. Added new Adroit dexterous manipulation domain in Table 1.
2. Included more descriptions of each environment and its observation space and added discussion of task complexity in Section 4.1 and Appendix A.
3. Revised the discussion of the extrapolation capability of the backwards model and design choice of the action selection strategies in Section 3.3.
4. Added more description of the forward dynamics baseline.
5. Reworded the problem setting in Section 3.2 to make the assumptions/limitations of our method clear.
6. Made the reversible state transition assumption more explicit.
7. Included additional experiment of training the model first and then the policy in Section 5.2 and Appendix.

---

### Meta-Review · Area_Chair_4BYS · 2022-08-26

**Recommendation:** Accept
**Confidence:** Less certain

**Metareview:**

The paper proposes training a backward model to teach agents to recover from drifting off the optimal state trajectories provided by a limited number of demonstrations. All reviewers have voted to (weakly) accept.

**Award:**

No

---

### Decision · Program_Chairs · 2022-09-14

Accept